# ILPO-NET: CONVOLUTION NETWORK FOR THE RECOGNITION OF ARBITRARY VOLUMETRIC PATTERNS

## ABSTRACT

Modern spatial data analysis is built on the effective recognition of spatial patterns and learning their hierarchy. Applications to real-world volumetric data require techniques that ensure invariance not only to shifts but also to pattern rotations. While traditional methods can readily achieve translational invariance, rotational invariance possesses multiple challenges and remains an active area of research. Here, we present ILPO-Net (Invariant to Local Patterns Orientation Network), a novel approach to handling arbitrarily shaped patterns with the convolutional operation inherently invariant to local spatial pattern orientations. Our architecture seamlessly integrates the new convolution operator and, when benchmarked on diverse volumetric datasets such as MedMNIST and CATH, demonstrates superior performance over the baselines with significantly reduced parameter counts—up to 1000 times fewer in the case of MedMNIST. Beyond these demonstrations, ILPO-Net's rotational invariance paves the way for other applications across multiple disciplines.

## 1    INTRODUCTION

In the constantly evolving world of data science, three-dimensional (3D) data models have emerged as a focal point of academic and industrial research. As the dimensionality of data extends beyond traditional 1D signals and 2D images, capturing the third dimension opens new scientific challenges and brings various opportunities. The possible applications of new methods range from sophisticated 3D models in computer graphics to the analysis of volumetric medical scans.

With the advent of deep learning, techniques that once revolutionized two-dimensional image processing are now being adapted and extended to deal with the volumetric nature of 3D data. However, the addition of the third dimension not only increases the computational complexity but also opens new theoretical challenges. One of the most pressing ones is the need for persistent treatment of volumetric data in arbitrary orientation. A particular example is medical imaging, where the alignment of a scan may vary depending on the equipment, the technician, or even the patient.

However, achieving such rotational consistency is non-trivial. While data augmentation techniques, such as artificially rotating training samples, can help to some extent, they do not inherently equip a neural network with the capability to recognize rotated patterns. Moreover, such methods can significantly increase the computational cost, especially with high-resolution 3D data. The community witnessed a spectrum of novel approaches specifically designed for these challenges. As we will see below, they range from modifications of traditional convolutional networks to the introduction of entirely new paradigms built on advanced mathematical principles.

## 2    RELATED WORK

Neural networks designed to process spatial data learn the data hierarchy by detecting local patterns and their relative position and orientation. However, when dealing with data in two or more dimensions, these patterns can be oriented arbitrarily, which makes neural network predictions dependent on the orientation of the input data. Classically, this dependence can be bypassed through data augmentation with rotated copies of data samples in the training set (Krizhevsky et al., 2012). In the volumetric (3D) case, augmentation often results in significant extra computational costs. For some

types of three-dimensional data, the canonical orientation of data samples or their local volumetric patterns can be uniquely defined (Pagès et al., 2019; Jumper et al., 2021; Igashov et al., 2021; Zhemchuzhnikov et al., 2022). In most real-world scenarios, though, it is common for 3D data to be oriented arbitrarily. Thus, there was a pressing need for methods with specific properties of rotational invariance or equivariance by design. We can trace two main directions in the development of these methods: those based on equivariant operations in the $SO(3)$ space (space of rotations in 3D), and those with learnable filters that are orthogonal to the $SO(3)$ or $SE(3)$ (roto-translational) groups.

The pioneering method from the first class was the Group Equivariant Convolutional Networks (G-CNNs) introduced by Cohen & Welling (2016), who proposed a general view on convolutions in different group spaces. Many more methods were built up subsequently upon this approach (Worrall & Brostow, 2018; Winkels & Cohen, 2022; Bekkers et al., 2018; Wang et al., 2019; Romero et al., 2020; Dehmamy et al., 2021; Roth & MacDonald, 2021; Knigge et al., 2022; Liu et al., 2022b). Several implementations of Group Equivariant Networks were specifically adapted for regular volumetric data, e.g., CubeNet(Worrall & Brostow, 2018) and 3D G-CNN (Winkels & Cohen, 2022). The authors of these methods consider a discrete set of 90-degree rotations and reflections, which exhaustively describe the possible positions of a cubic pattern on a regular grid. However, we shall note that, typically, both discrete and regular data are representations of the continuous realm, which embodies a continuous range of rotations. As a result, they cannot be limited to just a finite series of 90-degree turns. Another limitation is that this group of methods performs summing over rotations that can lead to the higher output of radially-symmetrical filters. This limits the expressiveness of the models because the angular dependencies of patterns are not memorized in the filters. Another branch in this development direction was represented by methods aimed at detecting patterns on a sphere. In Spherical CNNs, Cohen et al. proposed a convolution operation defined on the spherical surface, making it inherently rotationally equivariant (Cohen et al., 2018). Spherical CNNs are a comprehensive tool for working with spherical data, but they have limited application to volumetric cases. Even if we consider the 3D data as a collection of spheres, the application of this method will not be geometrically expressive since it will not provide translational invariance. Indeed, the current implementations of the method use spheres centered at the same point and thus will be sensitive to shifts in the input data. When thinking of expanding this approach for volumetric data where each voxel possesses its own coordinate system, there remains the challenge of information exchange between different spheres.

Let us characterize methods from the second class without delving deeply into mathematical terms. Here, each layer of the network operates with products of pairs of oriented input quantities. These products inherit the orientation of the input, and then they are summed up with learnable weights. The first two methods to be mentioned in this section are the Tensor Field Networks (TFN) (Thomas et al., 2018) and the N-Body Networks (NBNs) (Kondor, 2018). Kondor et al. (2018) presented a similar approach, the Clebsch-Gordan Nets applied to data on a sphere. These models employed spherical tensor algebra working on irregular point clouds. Weiler et al. (2018) proposed 3D Steerable CNNs, where the same algebra was applied to regular voxelized data. These three methods share the convolution operation having limited expressiveness that is shown in Appendix A. Satorras et al. (2021) built their approach on the same idea. However, they achieved the equivariance in a much simpler way without the usage of Clebsch-Gordan coefficients and spherical harmonics. Even though the authors handled pairwise interactions of elements in a point cloud, they still adopt an approach where equivariance is achieved by using equivariant quantities by design, which prevents them from capturing complex patterns involving multiple points. It is also worth mentioning the work of Ruhe et al. (2023), where the authors applied the same logic to geometric algebra but used multivectors instead of irreducible representations of $S^2$.

Apart from the two main directions, we can highlight the application of differential geometry, such as moving frames, to volumetric data, as demonstrated by Sangalli et al. (2023). This approach uses local geometry to set up the local pattern orientation. This idea unites the method with the family of Gauge networks (Cohen et al., 2019). The current implementation still depends on the discretization of input data. Rotating input samples can significantly reduce accuracy, as shown in (Sangalli et al., 2023).

Considering the points mentioned above, there is a need to create a technique that can detect local patterns of any shape in input 3D data, regardless of their orientation. This method should approach spaces $\mathbb{R}^3$ and $SO(3)$ differently. While operating in $\mathbb{R}^3$ requires a convolution, summation over

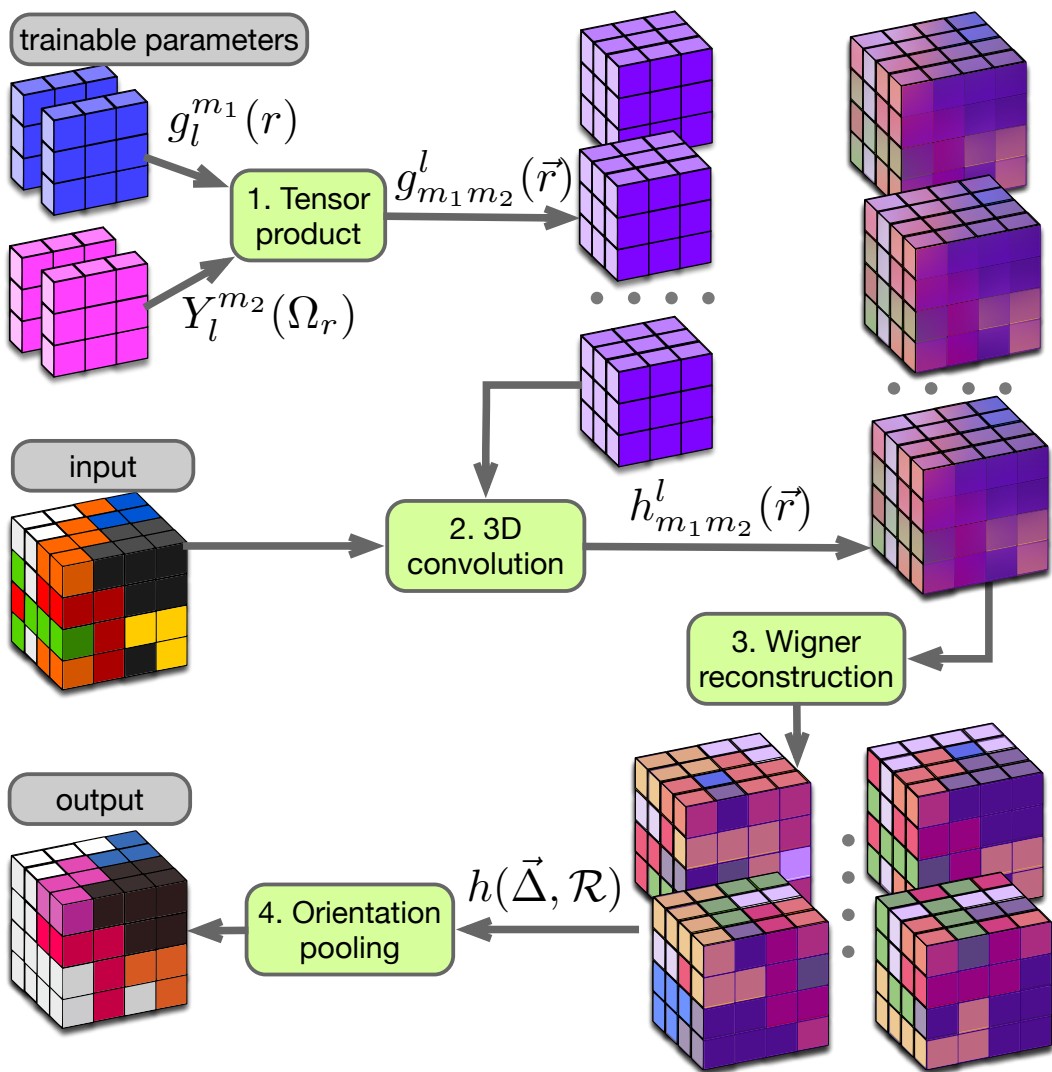

Figure 1: Schematic illustration of the ILPO convolution. The diagram showcases the main steps involved in our convolution process.

orientations in the rotational space should be avoided. Andrearczyk et al. followed this approach in (Andrearczyk et al., 2020), but they restricted the shape of the learnable filters. Additionally, their method has a narrow application domain, whereas we intend to develop a data-generic technique. Below, we propose a novel convolutional operation, Invariant to Local Features Orientation Network layer, that can detect arbitrary volumetric patterns, regardless of their orientations. This operation can be used in any convolutional architecture without major modifications. Our experiments on several datasets, CATH and the MedMNIST collection, demonstrate that this operation can achieve higher accuracy than the state-of-the-art methods with up to 3 orders of magnitude fewer learnable parameters.

# 3 THEORY

## 3.1 PROBLEM STATEMENT

The conventional 3D convolution can be formally expressed as:

$$h(\vec{\Delta}) = \int_{\mathbb{R}^3} f(\vec{r} + \vec{\Delta})g(\vec{r})d\vec{r}, \tag{1}$$

where $f(\vec{r})$ is a function describing the input data, $g(\vec{r})$ is a filter function, and $h(\vec{\Delta})$ is the convolution output function that depends on the position of the filter with respect to the original data $\vec{\Delta}$. The meaning of this operation in light of pattern recognition is that the value of the overlap integral of the filter and the fragment of the input data map around point $\vec{\Delta}$ serves as an indicator of the presence of the pattern in this point. However, such a recognition works correctly only if the orientation of the pattern in the filter and in the input data coincide. Therefore, if the applied pattern has a wrong orientation, a conventional convolution operation cannot recognize it.

The logical solution would be to apply the filter in multiple orientations. Then, the orientation of the filter appears in the arguments of the output function. In this approach, we consider a convolution with a rotated filter, represented as $g(\mathcal{R}\vec{r})$, where $\mathcal{R} \in SO(3)$,

$$h(\vec{\Delta}, \mathcal{R}) = \int_{\mathbb{R}^3} f(\vec{r} + \vec{\Delta})g(\mathcal{R}\vec{r})d\vec{r}. \tag{2}$$

The outcome of this convolution depends on both the shift $\vec{\Delta}$, and the filter rotation $\mathcal{R}$. The output function $h(\vec{r}, \mathcal{R})$ is now defined in $6D$ but if we want to obtain a 3D map that indicates that a pattern $g(\vec{r})$ in arbitrary orientation was detected at a point $\vec{\Delta}$ of map $f(\vec{r} + \vec{\Delta})$, we need to conduct an additional *orientation pooling* operation:

$$h(\vec{\Delta}) = \text{OrientionPool}_{\mathcal{R}}[h(\vec{\Delta}, \mathcal{R})], \tag{3}$$

which can generally be defined in different ways. The only constraint on this operation is that it must be *rotationally invariant* with respect to $\mathcal{R}$ or, in the discrete case, *invariant to the permutation* of the set of rotations:

$$\text{OrientionPool}_{\mathcal{R}}[f(\mathcal{R})] = \text{OrientionPool}_{\mathcal{R}}[f(\mathcal{R}\mathcal{R}')] \ \forall f \ \text{and} \ \forall \mathcal{R}'. \tag{4}$$

The simplest pooling operation satisfying this constraint would be an average over orientations $\mathcal{R}$. However, this will be equivalent to averaging the filter $g(\vec{r})$ over all possible orientations. Such an averaged filter is radially symmetric and is thus not very expressive. A better $\text{OrientionPool}_{\mathcal{R}}$ operation would be extracting a maximum over orientations $\mathcal{R}$ or applying a softmax operation, as defined below,

$$\begin{aligned}
\max_{\mathcal{R}} f(\mathcal{R}) &= \lim_{n\to\infty} \sqrt[n]{\int_{\text{SO(3)}} f^n(\mathcal{R})d\mathcal{R}} \\
\text{softmax}_{\mathcal{R}} f(\mathcal{R}) &= \frac{\int_{\text{SO(3)}} \text{relu}(f(\mathcal{R}))^2 d\mathcal{R}}{\int_{\text{SO(3)}} \text{relu}(f(\mathcal{R}))d\mathcal{R}}
\end{aligned} \tag{5}$$

Attempting to incorporate such a convolution in a neural network, we face several challenges.

1. If we assume $g(\vec{r})$ to be a learnable filter, it is not trivial to guarantee the correct back-propagation from multiple orientations of the filter to the original orientation of the filter $g(\vec{r})$.

2. Finding the hard- or soft- maximum in the pooling operation in the discrete case requires a consideration of a large number of rotations in the SO(3) space to reduce the deviation of the sampling maximum from the true maximum. To make the method feasible we need to avoid performing the 3D convolution for each of these rotations.

## 3.2 METHOD

Any square-integrable function on a unit sphere $g(\Omega) : S^2 \to \mathbb{R}$ can be expanded as a linear combination of spherical harmonics $Y_l^m(\Omega)$ of degrees $l$ and orders $k$ as

$$g(\Omega) = \sum_{l=0}^{\infty} \sum_{m=-l}^{l} g_l^m Y_l^m(\Omega). \tag{6}$$

The expansion coefficients $f_l^m$ can then be obtained by the following integrals,

$$g_l^m = \int_{S^2} g(\Omega) Y_l^m(\Omega) d\Omega. \tag{7}$$

Wigner matrices $D_{m_1 m_2}^l(\mathcal{R})$ are defined for $\mathcal{R} \in SO(3)$ and provide a representation of the rotation group $SO(3)$ in the space of spherical harmonics:

$$Y_l^{m_1}(\mathcal{R}\Omega) = \sum_{m_2=-l}^{l} D_{m_1 m_2}^l(\mathcal{R}) Y_l^{m_2}(\Omega). \tag{8}$$

Since Wigner matrices are orthogonal, i.e.,

$$\int_{SO(3)} D_{m_1 m_2}^l(\mathcal{R}) D_{k_1' k_2'}^{l'}(\mathcal{R}) d\mathcal{R} = \frac{8\pi^2}{2l+1} \delta_{ll'} \delta_{m_1 m_1'} \delta_{m_2 m_2'}, \tag{9}$$

any square-integrable function $h(\mathcal{R}) \in L^2(SO(3))$ can be decomposed into them as

$$h(\mathcal{R}) = \sum_{l=0}^{\infty} \sum_{m_1=-l}^{l} \sum_{m_2=-l}^{l} h_{m_1 m_2}^l D_{m_1 m_2}^l(\mathcal{R}), \tag{10}$$

where the expansion coefficients $h_{m_1 m_2}^l$ are obtained by integration as

$$h_{m_1 m_2}^l = \frac{2l+1}{8\pi^2} \int_{SO(3)} h(\mathcal{R}) D_{m_1 m_2}^l(\mathcal{R}) d\mathcal{R}. \tag{11}$$

Let us now consider the following decomposition of a function $h(\vec{\Delta}, \mathcal{R})$,

$$h(\vec{\Delta}, \mathcal{R}) = \sum_{l, m_1, m_2} h_{m_1 m_2}^l(\vec{\Delta}) D_{m_1 m_2}^l(\mathcal{R}). \tag{12}$$

Inserting the spherical harmonics decomposition of the rotated kernel $g(\mathcal{R}\vec{r})$ in Eq. 2, we obtain

$$h(\vec{\Delta}, \mathcal{R}) = \int_{\mathbb{R}^3} f(\vec{r}+\vec{\Delta}) \sum_{lm_1} g_l^{m_1}(r) Y_l^{m_1}(\mathcal{R}\Omega_r) d\vec{r} = \int_{\mathbb{R}^3} f(\vec{r}+\vec{\Delta}) \sum_{lm_1} g_l^{m_1}(r) \sum_{m_2} D_{m_1 m_2}^l(\mathcal{R}) Y_l^{m_2}(\Omega_r) d\vec{r}. \tag{13}$$

Changing the order of operations, we get the following expression,

$$h(\vec{\Delta}, \mathcal{R}) = \sum_{lm_1 m_2} D_{m_1 m_2}^l(\mathcal{R}) \int_{\mathbb{R}^3} f(\vec{r}+\vec{\Delta}) g_{m_1 m_2}^l(\vec{r}) d\vec{r}, \tag{14}$$

where we introduce expansion coefficients $g_{m_1 m_2}^l(\vec{r})$ at a point $\vec{r}$ with the radial and angular components $(r, \Omega)$ as

$$g_{m_1 m_2}^l(\vec{r}) = g_l^{m_1}(r) Y_l^{m_2}(\Omega_r). \tag{15}$$

Consequently, equating Eq. 12 to Eq. 14 and applying orthogonal conditions from Eq. 9 on both sides, we obtain

$$h_{m_1 m_2}^l(\vec{\Delta}) = \int_{\mathbb{R}^3} f(\vec{\Delta}+\vec{r}) g_{m_1 m_2}^l(\vec{r}) d\vec{r}. \tag{16}$$

In summary, our method comprises four steps, as depicted in Figure 1:

1. **Tensor product** of $g_l^{m_1}(r)$ and $Y_l^{m_2}(\Omega_r)$, where $g_l^{m_1}(r)$ are learnable filters (see Eq. 15).
2. **3D convolution** involving $g_{m_1 m_2}^l(\vec{r})$ and $f(\vec{r})$, with $f(\vec{r})$ representing the input data (refer to Eq 16).
3. **Wigner reconstruction** of $h(\vec{\Delta}, \mathcal{R})$ (see Eq. 12).
4. **Orientation pooling** as detailed in Eq. 5.

By employing these steps, we reduce the computational complexity through the utilization of Wigner matrices following the 3D convolution. The connection between the number the sampled points in $SO(3)$ and the number of coefficients is elaborated upon in Appendix C. Furthermore, subsection 4.1 presents an empirical examination of how these quantities influence the maximum sampling error. Appendix B provides details of the implementation of the method in the discrete case.

## 4 RESULTS

### 4.1 ORIENTATION INVARIANCE

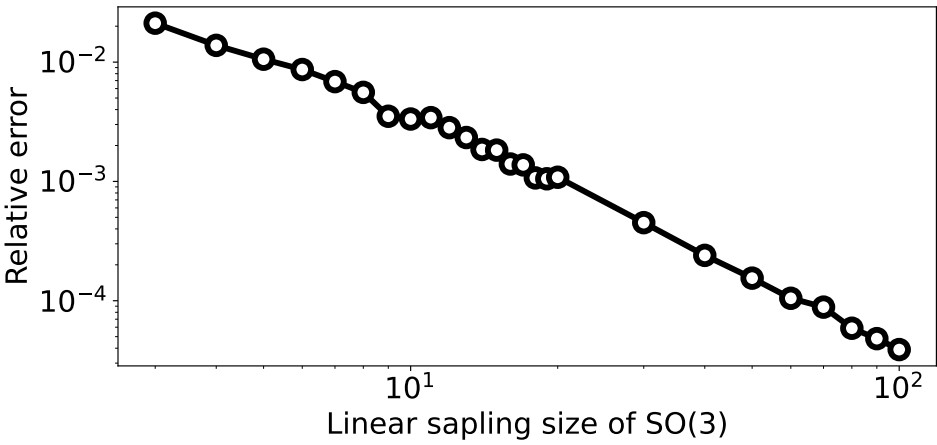

Figure 2: Standard deviation of sampled maxima relative to the true function maximum as a function of sampling size $K$ in SO(3) space.

To investigate the sensitivity of orientation-independent pattern detection to the linear size of sampling, we conducted the following experiment. We initiate a function in the $SO(3)$ space with a Wigner matrix decomposition up to a maximum degree of 2 ($L = 3$). This function is initiated by random generation of its decomposition coefficients. To probe the function's behavior under various orientations, we applied 100 random rotations to it, producing a collection of rotated copies. For each of these rotated versions, we found its sampled maximum over the $SO(3)$ space with the sampling size $K$. Aggregating these maxima across all rotations allowed us to determine their standard deviation.

Figure 2 shows the normalized standard deviation (relative to the true maximum of the initial function) as a function of the linear sampling size $K$. Even for relatively small values of $K = L$, the ratio between the standard deviation of the maxima and the true maximum hovers around $10^{-2}$. This implies that the deviation of the sampling maximum from the true maximum remains minimal, underscoring the reliability of our orientation-independent pattern detection across varying sampling resolutions.

### 4.2 EXPERIMENTS ON THE CATH DATASET

For our first experiment, we chose a volumetric voxelized dataset from (Weiler et al., 2018) composed of 3D protein conformations classified according to the CATH hierarcy. The CATH Protein Structure Classification Database provides a hierarchical classification of 3D conformations of protein domains, i.e., compact self-stabilizing protein regions that folds independently (Knudsen & Wiuf, 2010). The dataset considers the "architecture" level in the CATH hierarchy, version 4.2 (see `http://cathdb.info/browse/tree`). It focuses on "architectures" with a minimum of 700 members, producing ten distinct classes. All classes are represented by the same number of proteins. Each protein in the dataset is described by its alpha-carbon positions that are placed on the volumetric grid of the linear size 50. The dataset is available at `https://github.com/wouterboomsma/cath_datasets` (Weiler et al., 2018). For benchmarking, the authors of the dataset also provide a 10-fold split ensuring the variability of proteins from different splits.

For the experiment, we constructed three architectures (ILPONet, ILPONet-small, and ILPONet-tiny) with different numbers of trainable parameters, and also tested the two types of pooling op-

erations. ILPONet, ILPONet-small, and ILPONet-tiny replicate the architecture of ResNet-34 (He et al., 2016), but they impliment the novel convolution operation with 4, 8, and 16 times fewer channels on each layer, respectively. We conducted experiments for two types of orientation pooling with $K = 4$ for the soft-max version, and $K = 7$ for the hardmax version.

We compared the performance of ILPO-Net (our method) with two baselines: ResNet-34 and its equivariant version, ResNet-34 with Steerable filters, whose performance was demonstrated in (Weiler et al., 2018) where the dataset was introduced. Table 1 lists the accuracy (**ACC**) and the number of parameters(**# of params**) of different tested methods. Since the classes in the dataset are balanced, we can use accuracy as the sole metric to evaluate the precision of predictions.

| Method | ACC | # of params |
|---|---|---|
| ResNet-34 | 0.61 | 15M |
| Steerable ResNet-34 | 0.66 | 150K |
| ILPONet-34(hardmax) | **0.74** | 1M |
| ILPONet-34(softmax) | 0.74 | 1M |
| ILPONet-34(hardmax)-small | 0.73 | 258k |
| ILPONet-34(softmax)-small | 0.72 | 258k |
| ILPONet-34(hardmax)-tiny | 0.68 | 65k |
| ILPONet-34(softmax)-tiny | 0.70 | 65k |

Table 1: Performance comparison of various methods on the CATH dataset.

As shown in Table 1, all versions of ILPO-Net outperform both baselines on the CATH dataset. Furthermore, when comparing the number of parameters, even the smallest variant of ILPO-Net achieves a better accuracy, while having substantially fewer parameters than the equivariant baseline, Steerable Network.

**Technical details:** We used the first 7 splits for training, 1 for validation, and 2 for testing following the protocol of Weiler et al. (2018). We trained our models for 100 epochs with the Adam optimizer (Kingma & Ba, 2014) and an exponential learning rate decay of $0.94$ per epoch starting after an initial burn-in phase of 40 epochs. We used a $0.01$ dropout rate, and $L1$ and $L2$ regularization values of $10^{-7}$. For the final model, we chose the epoch where the validation accuracy was the highest. Table 1 shows the performance on the test data. We based our experiments on the framework provided by Weiler et al. (2018) in their **se3cnn** repository. We introduced our ILPO operator into the provided setup for training and evaluation.

### 4.3 EXPERIMENTS ON MEDMNIST DATASETS

For the second experiment, we selected MedMNIST v2, a vast MNIST-like collection of standardized biomedical images (Yang et al., 2023). This collection covers 12 datasets for 2D and 6 datasets for 3D images. Preprocessing reduced all images into the standard size of $28 \times 28$ for 2D and $28 \times 28 \times 28$ for 3D, each with its corresponding classification labels. MedMNIST v2 data are supplied with tasks ranging from binary/multi-class classification to ordinal regression and multi-label classification. The collection, in total, consists of 708,069 2D images and 9,998 3D images. For this study, we focused only on the 3D datasets of MedMNIST v2.

As the baseline, we used the same models as the authors of the collection tested on 3D datasets. These are multiple versions of ResNet (He et al., 2016) with 2.5D/3D/ACS (Yang et al., 2021) convolutions and open-source AutoML tools, auto-sklearn (Feurer et al., 2019), AutoKeras (Jin et al., 2019), FPVT (Liu et al., 2022a), and Moving Frame Net (Sangalli et al., 2023). As in the previous experiment, we constructed and trained multiple architectures (ILPONet, and ILPONet-small) of different size with two versions of the orientation pooling operation. They repeat the sequence of layers in ResNet-18 and ResNet-50 but they do not reduce the size of the spatial input dimension throughout the network.

The models *ILPONet-small* and *ILPONet* keep 4 and 8 feature channels, respectively, throughout the network. We tested these architectures for both soft- and hardmax orientation pooling strategies. Table 2 lists the performance of our models compared to the baselines. Here, the classes are not balanced. Therefore, the accuracy (**ACC**) cannot be the only indicator of the prediction precision,

and we also consider AUC-ROC(**AUC**) that is more revealing. We can see that ILPOResNet models, even with a substantially reduced number of parameters, demonstrate competitive or superior performance compared to traditional methods on the 3D datasets of MedMNIST v2.

**Technical details:** For each dataset, we used the training-validation-test split provided by Yang et al. (2023). We utilized the Adam optimizer with an initial learning rate of $0.0005$ and trained the model for $100$ epochs, delaying the learning rate by $0.1$ after $50$ and $75$ epochs. The dropout rate was $0.01$. To test the model we choose the epoch corresponding to the best **AUC** on the validation set. We based our experiments on the framework provided by Yang et al. (2023) in their **MedMNIST** repository. We introduced our ILPO operator into their setup for training and evaluation.

| Methods | # of params | Organ | | Nodule | | Fracture | | Adrenal | | Vessel | | Synapse | |
|---|---|---|---|---|---|---|---|---|---|---|---|---|---|
| | | AUC | ACC | AUC | ACC | AUC | ACC | AUC | ACC | AUC | ACC | AUC | ACC |
| ResNet-18 + 2.5D | 11M | 0.977 | 0.788 | 0.838 | 0.835 | 0.587 | 0.451 | 0.718 | 0.772 | 0.748 | 0.846 | 0.634 | 0.696 |
| ResNet-18 + 3D | 33M | **0.996** | **0.907** | 0.863 | 0.844 | 0.712 | 0.508 | 0.827 | 0.721 | 0.874 | 0.877 | 0.820 | 0.745 |
| ResNet-18 + ACS | 11M | 0.994 | 0.900 | 0.873 | 0.847 | 0.714 | 0.497 | 0.839 | 0.754 | 0.930 | 0.928 | 0.705 | 0.722 |
| ResNet-50 + 2.5D | 15M | 0.974 | 0.769 | 0.835 | 0.848 | 0.552 | 0.397 | 0.732 | 0.763 | 0.751 | 0.877 | 0.669 | 0.735 |
| ResNet-50 + 3D | 44M | 0.994 | 0.883 | 0.875 | 0.847 | 0.725 | 0.494 | 0.828 | 0.745 | 0.907 | 0.918 | 0.851 | 0.795 |
| ResNet-50 + ACS | 15M | 0.994 | 0.889 | 0.886 | 0.841 | 0.750 | 0.517 | 0.828 | 0.758 | 0.912 | 0.858 | 0.719 | 0.709 |
| auto-sklearn* | - | 0.977 | 0.814 | **0.914** | **0.874** | 0.628 | 0.453 | 0.828 | 0.802 | 0.910 | 0.915 | 0.631 | 0.730 |
| AutoKeras* | - | 0.979 | 0.804 | 0.844 | 0.834 | 0.642 | 0.458 | 0.804 | 0.705 | 0.773 | 0.894 | 0.538 | 0.724 |
| FPVT* | - | 0.923 | 0.800 | 0.814 | 0.822 | 0.640 | 0.438 | 0.801 | 0.704 | 0.770 | 0.888 | 0.530 | 0.712 |
| SE3MovFrNet * | - | - | 0.745 | - | 0.871 | - | 0.615 | - | 0.815 | - | **0.953** | - | 0.896 |
| ILPOResNet-18(softmax)-small | 7k | 0.960 | 0.631 | 0.887 | 0.848 | 0.791 | 0.579 | 0.897 | 0.805 | 0.815 | 0.838 | 0.804 | 0.517 |
| ILPOResNet-18(hardmax)-small | 7k | 0.951 | 0.600 | 0.906 | 0.861 | **0.808** | **0.642** | 0.870 | 0.792 | 0.925 | 0.908 | 0.825 | 0.750 |
| ILPOResNet-18(softmax) | 29k | 0.967 | 0.716 | 0.894 | 0.871 | 0.761 | 0.558 | **0.910** | **0.856** | 0.908 | 0.919 | 0.836 | 0.815 |
| ILPOResNet-18(hardmax) | 29k | 0.971 | 0.705 | 0.900 | **0.874** | 0.773 | 0.580 | 0.897 | 0.846 | 0.927 | 0.908 | 0.800 | 0.767 |
| ILPOResNet-50(softmax)-small | 10k | 0.979 | 0.757 | 0.902 | 0.865 | 0.772 | 0.558 | 0.864 | 0.745 | 0.864 | 0.890 | 0.880 | 0.844 |
| ILPOResNet-50(hardmax)-small | 10k | 0.981 | 0.780 | 0.887 | 0.861 | 0.768 | 0.571 | 0.841 | 0.792 | **0.937** | 0.901 | 0.861 | 0.784 |
| ILPOResNet-50(softmax) | 38k | 0.992 | 0.879 | 0.912 | 0.871 | 0.767 | 0.608 | 0.869 | 0.809 | 0.829 | 0.851 | **0.940** | **0.923** |
| ILPOResNet-50(hardmax) | 38k | 0.975 | 0.754 | 0.911 | 0.839 | 0.769 | 0.521 | 0.893 | 0.842 | 0.902 | 0.885 | 0.885 | 0.858 |

Table 2: Comparison of different methods on MedMNIST's 3D datasets. ($^*$) For these methods, the number of parameters is unknown.

## 4.4 FILTER DEMONSTRATION

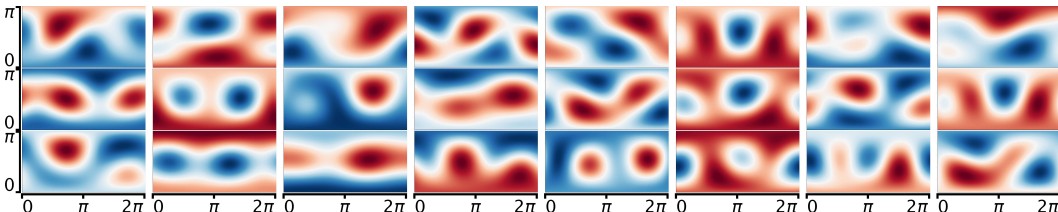

Figure 3: Visualization of filters from the 1st ILPO layer of ILPONet-50. Each column corresponds to different output channels, with rows indicating different radii and input channels. Given that the first ILPO layer only has one input channel, only three projections (radii) are shown in each column. $x$ and $y$ axes correspond to the azimuthal and polar angles, correspondingly. The filters' values are shown in the Mercader projection. The red color corresponds to the positive values, and the blue color to the negative ones.

For a deeper understanding of our models, it is useful to delve into the visualizations of their filters. Of the numerous experiments conducted, we opted to focus on the MedMNIST experiments, primarily due to the smaller size of the trained models in terms of parameter count. Within the MedMNIST collection, we chose the Synapse dataset because of its more sophisticated and variable patterns, and analysed the filters from the top-performing ILPONet-50 model with the softmax orientation pooling. This architecture employs ILPO convolutional layers, each having a filter size of $L = 3$. Here, we demonstrate filters from the first and the last ILPO layers. Depending on the radius $(r)$, these filters could represent a single point (for $r = 0$) or spheres for other radii values. We use the Mercader projection to show values on the filters' spheres for $r > 0$, in two spherical angles, azimuthal and polar.

Figure 3 shows the first ILPO layer. The layer has a single input channel. Different rows correspond to different radii($r = 1, \sqrt{2}, \sqrt{3}$), whereas each column corresponds to a different output channel. Appendix D also shows the 17th ILPO layer.

These figures demonstrate a variety of memorized patterns. We can see no spatial symmetry in the filters and that the presented model is capable of learning filters of arbitrary shape. Interestingly, we cannot spot a clear difference between the filters of the first and the last layers.

## 5    DISCUSSION AND CONCLUSION

In real-world scenarios, data augmentation is commonly employed to achieve rotational and other invariances of DL models. While this method may significantly increase the dataset's size and the number of parameters, it can also limit the expressivity and explainability of the obtained models. Using invariant methods by design is a valid alternative that ensures consistent neural network performance.

We can adapt our approach to other representations beyond regular voxelized grids. For example, we can handle point clouds, though this would require the integration of a proper radial basis. If we are dealing with molecular data, we will also have to rethink the interactions among data elements (i.e., cloud points) – a step that we have currently skipped.

To conclude, we proposed the ILPO-NET approach that efficiently manages arbitrarily shaped patterns, providing inherent invariance to local spatial pattern orientations through the novel convolution operation. When tested against several volumetric datasets, ILPO-Net demonstrated state-of-the-art performance with a remarkable reduction in parameter counts. Its potential extends beyond the tested cases, with promising applications across multiple disciplines.

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

APPENDIX

## A    LIMITATION OF EXPRESSIVENESS IN STEERABLE NETWORKS

Convolution operations, foundational to modern Convolutional Neural Networks (CNNs), serve as a mechanism for detecting patterns in input data. In the traditional convolution, higher activation values in the feature map indicate regions in the input where there is a significant match with the convolutional filter, thereby signaling the presence of a targeted pattern.

Let us consider how this mechanism works in convolutions with steerable filters (Weiler et al., 2018). The steerable filter that maps between irreducible features ($i \to l$) is defined as:

$$\kappa_{il}(\vec{r}) = \sum_{L=|i-l|}^{i+l} \sum_{n=0}^{N-1} w_{il,Ln} \kappa_{il,Ln}(\vec{r}), \tag{17}$$

where $\kappa_{il}(\vec{r}) : \mathbb{R}^3 \to \mathbb{R}^{(2i+1)(2l+1)}$. Here, $w_{il,Ln}$ are learnable weights and $\kappa_{il,Ln}$ are basis functions given by:

$$\kappa_{il,Ln}(\vec{r}) = Q^{ilL} \eta_{Ln}(\vec{r}), \tag{18}$$

where $Q^{ilL} \in \mathbb{R}^{(2i+1)(2l+1)\times(2L+1)}$ is the 3-dimensional tensor with Clebsch-Gordon coefficients and

$$\eta_{Ln}(\vec{r}) = \phi_n(r) Y_L(\Omega_r), \tag{19}$$

$\eta_{Ln}(\vec{r}) : \mathbb{R}^3 \to \mathbb{R}^{(2L+1)}$ and $Y_L(\Omega_r)$ is a vector with spherical harmonics of degree $L$. Functions $\phi_n(n = 0, ..., N-1)$ form a radial basis. For a scalar field as input data and considering the special case $l = 0$, the filter reduces to

$$\kappa_{i0}(\vec{r}) = \sum_{n=0}^{N-1} w_{i0,in} \phi_n(r) Y_i(\Omega_r), \tag{20}$$

where $\kappa_{i0}(\vec{r}) : \mathbb{R}^3 \to \mathbb{R}^{(2i+1)\times 1}$.

Let us apply the convolution to the following input function,

$$f(\vec{r}) = \sum_{i=0}^{L_{\max}} \sum_{m=-i}^{i} \sum_{n=0}^{N} f_{in}^m \phi_n(r) Y_i^m(\Omega_r), \tag{21}$$

where indices $i, m$ correspond to the angular decomposition and $n$ is a radial index. Without loss of generality for the final conclusion, let us consider a special case when coefficients $f_{in}^m$ can be expressed as a product: $f_{in}^m = f_i^m q_{in}$. We also assume that the pattern presented by this function is localised and the function is defined in a cube. The filter $\kappa_{i0}(\vec{r})$ is localised in a cube of the same size. If we use the integral formulation, the result of the convolution operation at the center of the pattern will be:

$$h_i^m = \int_0^\infty \int_{S^2} f(\vec{r})[\kappa_{i0}(\vec{r})]_{m0} d\Omega_r r^2 dr = f_i^m \sum_{n=0}^{N-1} w_{i0,in} q_{in}. \tag{22}$$

Then, according to the logic of the convolution layer, a nonlinear operator is applied to the convolution result, which zeros the low signal level. Let us consider two types of nonlinearities used in 3D Steerable networks: gated- and norm-nonlinearity. In these operators, the high-degree output of the convolution result ($h_i^m, i > 0$) is multiplied with $\sigma(h_0^0)$ and $\sigma(\|h_i\|)$, respectively, where $\sigma$ is an activation function and $\|h_i\| = \sqrt{\sum_{m=-i}^{i}(h_i^m)^2} = |\sum_{n=0}^{N-1} w_{i0,in} q_{in}| \sqrt{\sum_{m=-i}^{i}(f_i^m)^2}$ is the norm of the $i$th-degree coefficients . In the first case, the gated non-linearity does not distinguish patterns of different shapes if they have the same decomposition coefficients of the 0th degree($f_{0n}^0$). The norm non-linearity brings more expressiveness for representations of the 1st-degree because if two sets of representations, $\{f_1^{-1}, f_1^0, f_1^1\}$ and $\{[f']_1^{-1}, [f']_1^0, [f']_1^1\}$, have equal norms ($\|f_1\| = \|[f']_1\|$), then $f_1$ can be retrieved from $[f']_1$ by a rotation or, in other words, they represent the same shape . However, this rule does not work for higher degrees ($i \geq 2$). For example, representations of the 2nd-degree $f_2 = \{1, 0, 0, 0, 0\}$ and $[f']_2 = \{0, 0, 1, 0, 0\}$ represent different shapes but have equal norms.

Accordingly, a single layer cannot cope with the recognition of an arbitrary pattern in the input data. Thus, the recognition task moves to the subsequent layers. However, on the second layer, there is an exchange between the voxel of the feature map where $h_i$ is stored and other voxels that contain not only the pattern information but also the pattern's neighbors information. Therefore, the result of the central pattern recognition will not be unique but depends on the pattern neighbors.

## B    IMPLEMENTATION FOR THE VOXELIZED DATA

### B.1    DISCRETE CONVOLUTION

Here we describe how the convolution introduced above can be discretized for use in a neural network with *regular voxelized data*. Let us firstly define for each filter $g(\vec{r})$, where $\vec{r} = (x_i, y_j, z_k)$, a regular Cartesian grid of a linear size $L$: $0 \leq i, j, k < L$. This size also defines the maximum expansion order of the spherical harmonics expansion in Eq. 6. Let us also compute spherical coordinates $(r_{ijk}, \Omega_{ijk})$ for a voxel with indices $i, j, k$ in the Cartesian grid with respect to the center of the filter. For each of data voxel of radii $r_{ijk}$ inside the filter grid, with the origin in the center of the grid, we define a filter $g^l_{m_1 m_2}(x_i, y_j, z_k)$ and parameterize it with learnable coefficients $g^m_l(r_{ijk})$ and non-learnable spherical harmonics basis functions according to Eq. 15 .

$$g^l_{m_1 m_2}(x_i, y_j, z_k) = g^{m_1}_l(r_{ijk}) Y^{m_2}_l(\Omega_{ijk}). \tag{23}$$

After, we conduct a *discrete* version of the 3D convolution from Eq. 16:

$$h^l_{m_1 m_2}(x_i, y_j, z_k) = \sum_{i'=0}^{L-1} \sum_{j'=0}^{L-1} \sum_{z'=0}^{L-1} f(x_{i+i'-L//2}, y_{j+j'-L//2}, z_{k+k'-L//2}) g^l_{m_1 m_2}(x_{i'}, y_{j'}, z_{k'}),$$
$$\tag{24}$$

where $f(x_i, y_j, z_k)$ is the input voxelized data. This operation has a complexity of $O(N^3 \times D_{\text{in}} \times D_{\text{out}} \times L^6)$, where the multiplier $L^6$ is composed of the size of the filter, $L^3$, and the number of $g^l_{m_1 m_2}$ coefficients $\propto L^3$ , $N$ is the linear size of the input data $f(\vec{r})$ and $D_{in}$ and $D_{out}$ are the number of the input and the output channels, respectively. For the computational efficiency of our method, we always keep the value of $L$ fixed and small, independent of $N$.

To perform the Wigner matrix reconstruction in Eq. 10, we need to numerically integrate the $SO(3)$ space. We can compute this integral *exactly* using the Gauss-Legendre quadrature scheme from $L$ points (Khalid et al., 2016). It is convenient to represent a rotation in $SO(3)$ by a successive application of three Euler angles $\alpha$, $\beta$ and $\gamma$, about the axes Z, Y and Z, respectively. Then, the Wigner matrix $D^l_{m_1 m_2}(\mathcal{R})$ can be expressed as a function of three angles: $D^l_{m_1 m_2}(\mathcal{R}) = D^l_{m_1 m_2}(\alpha, \beta, \gamma)$ and written as a sum of two terms:

$$D^l_{m_1 m_2}(\alpha, \beta, \gamma) = C_{m_1}(m_1 \alpha)[d_1]^l_{m_1 m_2}(\beta) C_{m_2}(m_2 \gamma) + C_{-m_1}(m_1 \alpha)[d_2]^l_{m_1 m_2}(\beta) C_{-m_2}(m_2 \gamma),$$
$$\tag{25}$$

where $[d_i]^l_{m_1 m_2}, i = 1, 2$ can be decomposed into associated Legendre polynomials $P^m_l(\cos(\beta)), 0 \leq m < l$ , and $C_m$ is defined as follows:

$$C_m(x) = \begin{cases} \cos(x), & m \geq 0 \\ \sin(x), & m < 0 \end{cases}. \tag{26}$$

Given such a form of $D^l_{m_1 m_2}(\alpha, \beta, \gamma)$, we discretize the space of rotations $SO(3)$ as a 3D space with dimensions along the $\alpha, \beta$ and $\gamma$ angles. The dimensions $\alpha$ and $\gamma$ have a regular division. We use the Gauss–Legendre quadrature to discretize $\cos(\beta)$ to define the $\beta$ dimension. Then, we perform the discrete version of the summation in Eq. 12:

$$h(x_i, y_j, z_k, \alpha_q, \beta_r, \gamma_s) = \sum_{m_2=-l}^{l} C_{m_2}(m_2 \gamma_s) (\sum_{m_1=-l}^{l} C_{m_1}(m_1 \alpha_q) (\sum_{l=0}^{L-1} [d_1]^l_{m_1 m_2}(\beta_r) h^l_{m_1 m_2}(x_i, y_j, z_k))) +$$

$$\sum_{m_2=-l}^{l} C_{-m_2}(m_2 \gamma_s) (\sum_{m_1=-l}^{l} C_{-m_1}(m_1 \alpha_q) (\sum_{l=0}^{L-1} [d_2]^l_{m_1 m_2}(\beta_r) h^l_{m_1 m_2}(x_i, y_j, z_k))), \quad (27)$$

where $0 \leq q, r, s \leq K - 1$, $K$ is the linear size of the $SO(3)$ space discretization. If we assume that $L < K$, then the complexity of the reconstruction is $O(N^3 \times D_{\text{out}} \times K^3 \times L)$, where $N$ is the linear

size of the input data $f(\vec{r})$, and $D_{in}$ and $D_{out}$ are the number of the input and the output channels, respectively. We shall specifically note that this operation has a lower complexity compared to the case of Eq. 2, if the latter is calculated with a brute-force approach provided that the number of sampled points in the $SO(3)$ space $K^3 >> L^3$.

### B.2 ORIENTATION POOLING

For the orientation pooling operation, we have considered two nonlinear operations, hard maximum and soft maximum defined in Eq. 5. While only $L^3$ points in the $SO(3)$ space are sufficient to find the exact value of the integration of functions $h(x_i, y_j, z_k, \alpha, \beta, \gamma)$, many more points are required to approximate the integration of $\mathrm{relu}(h(x_i, y_j, z_k, \alpha, \beta, \gamma))^2$ or $h^n(x_i, y_j, z_k, \alpha, \beta, \gamma)$. There is not a closed-form dependency between $K$, $L$ and $\epsilon$, the error of discrete approximation of integrals in Eq. 5 on the grid of $K^3$ points. However, we need to ensure that the deviation of the sampling maximum from the real maximum for a given sampling division $K$ is bounded. For this purpose we introduce lemmas and theorems in Appendix C.

Theorem C.4 provides an upper bound for the error of the sampled maximum. The softmax is limited by the hard maximum value for the continuous and discrete cases, consequently the sampled softmax error is also bounded. We also deduced an *empirical* relationship between the error and parameters $L$ and $K$ for both operations. For example, for $L = 3$ the error of the softmax approximation follows the relation $\epsilon = 4K^{-3}$. Therefore, for $\epsilon = 0.1, 0.05$ or $0.01$ we need to consider $K = 4, 5$ or $7$, respectively. The error of the sampling hardmax is approximately $2.75K^{-2}$ if $L = 3$. It means that $K = 7, 9$ or $30$ will give $\epsilon = 0.1, 0.05$ or $0.01$ respectively.

The discrete calculation of the hard maximumum does not differ from the continuous case. The discrete form of the soft maximum operation has the following expression:

$$\mathrm{softmax}_{\mathcal{R}} f(x_i, y_j, z_k, \mathcal{R}) = \frac{\sum_{q,r,s} w_r \mathrm{relu}(h(x_i, y_j, z_k, \alpha_q, \beta_r, \gamma_s))^2}{\sum_{q,r,s} w_r \mathrm{relu}(h(x_i, y_j, z_k, \alpha_q, \beta_r, \gamma_s))}, \tag{28}$$

where $w_r$ are the Gauss–Legendre quadrature weights.

## C UPPER BOUND OF THE SAMPLING MAXIMUM ERROR

**Lemma C.1.** *Let $Y_l^k(\theta, \phi)$ be the spherical harmonic function of degree $l$ and order $k$. Then, the Lipschitz constant $L$ of $Y_l^k(\theta, \phi)$ is bounded by:*

$$L \leq \sqrt{l(l+1)}$$

*Proof.* Given the following differential relations:

$$\frac{\delta Y_l^k(\theta, \phi)}{\delta \theta} = k Y_l^{-k}(\theta, \phi) \tag{29}$$

$$\frac{\delta Y_l^k(\theta, \phi)}{\delta \phi} = Y_l^k(\theta, \phi) l \cot(\phi) - Y_{l-1}^k(\theta, \phi) \frac{\sqrt{(l-k)(l+k)}}{\sin(\phi)} \frac{2l+1}{2l-1}, \tag{30}$$

we can obtain the expression for the gradient of $Y_l^k(\theta, \phi)$ as:

$$\nabla Y_l^k = \left( \frac{\delta Y_l^k}{\delta \theta}, \frac{\delta Y_l^k}{\delta \phi} \right). \tag{31}$$

To determine the Lipschitz constant, we find the maximum magnitude of the gradient over the function's domain. Using the provided differential relations, the squared magnitude of the gradient is:

$$\|\nabla Y_l^k\|^2 = \left( k Y_l^{-k} \right)^2 + \left( Y_l^k l \cot(\phi) - Y_{l-1}^k \frac{\sqrt{(l-k)(l+k)}}{\sin(\phi)} \frac{2l+1}{2l-1} \right)^2. \tag{32}$$

Given that $\|k\| \leq l$, the term $k^2$ is bounded by $l^2$. The dominant term from the second expression is $l \cot(\phi)$, which in the worst case is proportional to $l^2$. Thus, the Lipschitz constant is bounded by the square root of the maximum term from the gradient's squared magnitude. This gives:

$$L \leq \sqrt{l(l+1)}. \tag{33}$$

$\square$

**Theorem C.2.** *The Lipschitz constant $L_D$ of the Wigner matrix element $D_{k_1 k_2}^l(\mathcal{R})$ is bounded by:*

$$L_D \leq 4\pi\sqrt{l(l+1)}$$

*Proof.* First, recall the expression for the Wigner matrix element:

$$D_{k_1 k_2}^l(\mathcal{R}) = \int_{SO(2)} Y_l^{k_1}(\mathcal{R}x) Y_l^{k_2}(x) \, dx, \tag{34}$$

where $x = x(\theta, \phi)$ is a solid angle, and $\mathcal{R}$ is a rotation in $SO(3)$. To determine the Lipschitz constant for the Wigner matrix element, we find the magnitude of its gradient with respect to $\mathcal{R}$. Using the chain rule:

$$\frac{\partial Y_l^k(\mathcal{R}x)}{\partial \mathcal{R}} = \frac{\partial Y_l^k(x)}{\partial x}(\mathcal{R}x)\frac{\partial(\mathcal{R}x)}{\partial \mathcal{R}}. \tag{35}$$

Given the lemma above, we know that the Lipschitz constant $L$ for the spherical harmonic $Y_l^k(\theta, \phi)$ is bounded by $\sqrt{l(l+1)}$. Thus,

$$\max \|\frac{\partial D_{k_1 k_2}^l(\mathcal{R})}{\partial \mathcal{R}}\| \leq 4\pi\sqrt{l(l+1)} \max Y_l^{k_2} \leq 4\pi\sqrt{l(l+1)}. \tag{36}$$

$\square$

**Theorem C.3.** *Let $f(\mathcal{R})$ be a function in $SO(3)$ whose maximum degree of Wigner matrices decomposition is $L-1$ and whose 2-norm is $C$. Then, the Lipschitz constant $L_f$ of $f$ is bounded by:*

$$L_f \leq 4\frac{C}{\sqrt{3}}L^{\frac{5}{2}}$$

*Proof.* Given the decomposition of the function $f$ in terms of Wigner matrices,

$$f(\mathcal{R}) = \sum_{l=0}^{L-1} \sum_{k_1=-l}^{l} \sum_{k_2=-l}^{l} f_{k_1 k_2}^l D_{k_1 k_2}^l(\mathcal{R}), \tag{37}$$

we also have the expression for the 2-norm squared of $f$,

$$\|f\|_2^2 = \sum_{l=0}^{L-1} \sum_{k_1=-l}^{l} \sum_{k_2=-l}^{l} \frac{8\pi^2}{2l+1} \|f_{k_1 k_2}^l\|^2 = C^2. \tag{38}$$

Knowing that

$$\left(\sum_{l=0}^{L-1} \sum_{k_1=-l}^{l} \sum_{k_2=-l}^{l} |f_{k_1 k_2}^l|\right)^2 \leq \frac{(4L^3-L)}{3}\left(\sum_{l=0}^{L-1} \sum_{k_1=-l}^{l} \sum_{k_2=-l}^{l} \|f_{k_1 k_2}^l\|^2\right), \tag{39}$$

we deduce:

$$\frac{8\pi^2}{2L-1}\left(\sum_{l=0}^{L-1} \sum_{k_1=-l}^{l} \sum_{k_2=-l}^{l} |f_{k_1 k_2}^l|\right)^2 \leq \frac{(4L^3-L)}{3}C^2. \tag{40}$$

From the previous expression it follows that:

$$\sum_{l=0}^{L-1} \sum_{k_1=-l}^{l} \sum_{k_2=-l}^{l} \|f_{k_1 k_2}^l\| \leq \sqrt{C^2\frac{(4L^3-L)}{3}\frac{2L-1}{8\pi^2}}. \tag{41}$$

Considering that the Lipschitz constant for $D_{k_1 k_2}^l(\mathcal{R})$ is $4\pi\sqrt{l(l+1)}$, the Lipschitz constant for $f(\mathcal{R})$ is bounded by the product of the maximum Lipschitz constant for the Wigner matrices and the maximum magnitude of the coefficients. Thus,

$$L_f \leq 4\pi\sqrt{C\frac{(4L^3 - L)}{3}\frac{2L-1}{8\pi^2}}\sqrt{(L-1)L} \leq 4\frac{C}{\sqrt{3}}L^{\frac{5}{2}}. \tag{42}$$

This concludes the proof. $\square$

**Theorem C.4.** *Let the function $f(\mathcal{R})$ be defined in SO(3) with its maximum degree of Wigner matrices decomposition being $L-1$:*

$$f(\mathcal{R}) = \sum_{l=0}^{L-1} \sum_{m_1=-l}^{l} \sum_{m_2=-l}^{l} f_{m_1 m_2}^l D_{m_1 m_2}^l(\mathcal{R}),$$

*with the 2-norm of this function $C < \infty$. Given a sampling $\alpha_{k_1} = k_1\frac{2\pi}{K}, k_1 = 0, ..., K, \beta_{k_2} = \arccos(x_{k_2})$ where $x_i$ are Gauss-Legendre quadrature points of $K$, and $\gamma_{k_3} = k_3\frac{2\pi}{K}, k_3 = 0, ..., K$, if $K > K_0$ where $K_0 = \frac{8\pi L^{\frac{5}{2}} C/\sqrt{3}}{\epsilon}$, then the discrepancy between the sampled maximum and the true maximum of $f$ over its domain is smaller than $\epsilon$.*

*Proof.* Using the Lipschitz constant from Theorem C.3, we get:

$$|f(\mathbf{u}) - f(\mathbf{v})| \leq 4\frac{C}{\sqrt{3}}L^{\frac{5}{2}}\|\mathbf{u} - \mathbf{v}\|. \tag{43}$$

The largest difference in successive sampled points in $\alpha$ and $\gamma$ will be :

$$\|\mathbf{u}_{\text{successive}} - \mathbf{v}_{\text{successive}}\| = \frac{2\pi}{K}. \tag{44}$$

For the sampling in $\beta$ we obtain the same relation,

$$\max_i |\beta_{i+1} - \beta_i| \leq \frac{2\pi}{K}. \tag{45}$$

Using the Lipschitz property and combining the discrepancies, we deduce:

$$|f(\mathbf{u}_{\text{successive}}) - f(\mathbf{v}_{\text{successive}})| \leq 4\frac{C}{\sqrt{3}}L^{\frac{5}{2}}\frac{2\pi}{K}. \tag{46}$$

For the above discrepancy to be smaller than $\epsilon$, we must require:

$$K > \frac{8\pi L^{\frac{5}{2}} C/\sqrt{3}}{\epsilon}. \tag{47}$$

Thus, the smallest such a value for $K$ is $K_0 = \frac{8\pi L^{\frac{5}{2}} C/\sqrt{3}}{\epsilon}$. $\square$

## D   FILTER DEMONSTRATION

Figure 4 visualizes the last, 17th ILPO layer from the ILPONet-50 model trained on the Synapse dataset of the MedMNIST collection. This layer has multiple input channels, therefore we split each column into triplets corresponding to different input channels.

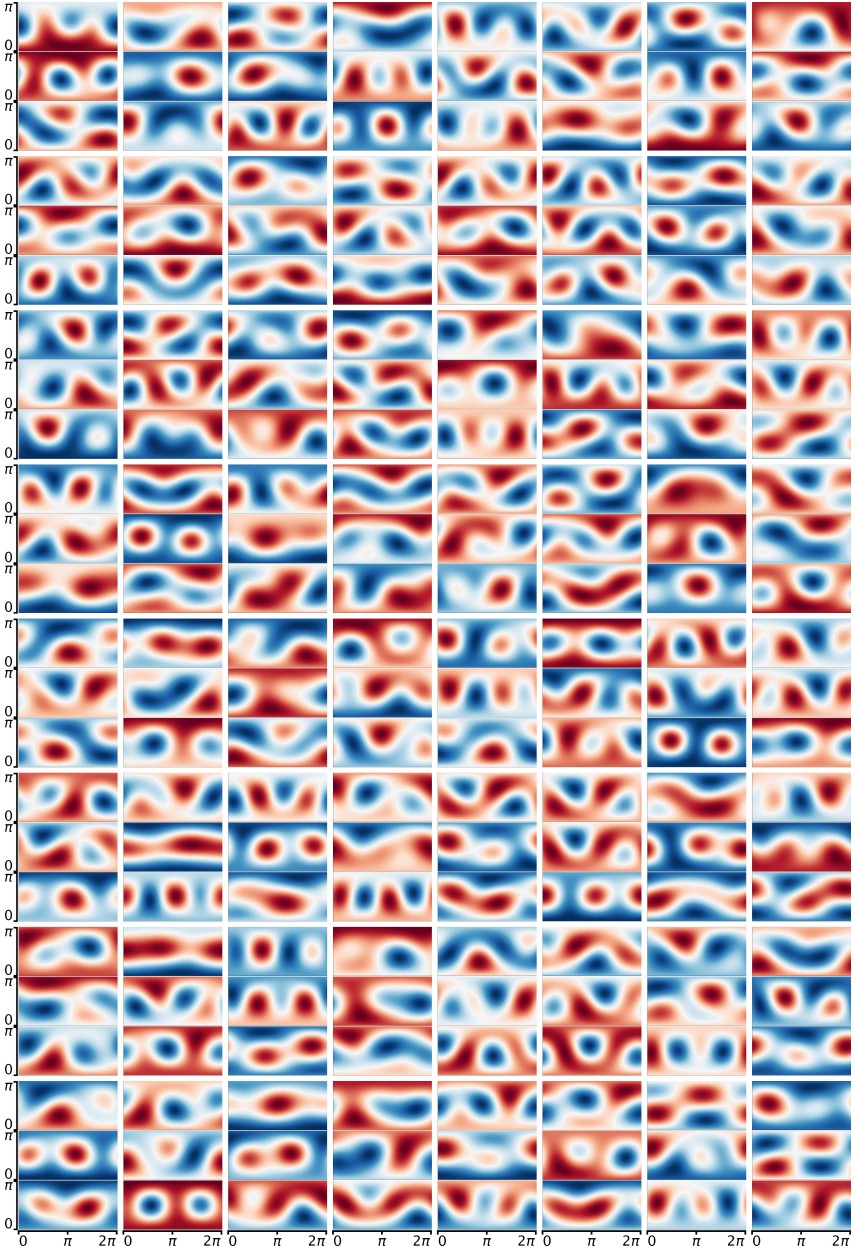

Figure 4: Visualization of filters from the last, 17th ILFO layer of ILFONet-50. Each column in the illustration represents a triplet corresponding to three different radii in the filter. Different triplets relate to different input channels, reflecting the complexity and feature extraction capabilities of deeper layers in the network. $x$ and $y$ axes correspond to the azimuthal and polar angles, correspondingly. The filters' values are shown in the Mercader projection. The red color corresponds to the positive values, and the blue color to the negative ones.

