# OpenReview forum: "ILPO-NET: convolution network for the recognition of arbitrary volumetric patterns"
_ICLR.cc/2024/Conference — Submitted to ICLR 2024_

### Official Review · Reviewer_ckuL · 2023-10-31

**Soundness:** 3 good
**Presentation:** 2 fair
**Contribution:** 1 poor
**Rating:** 3
**Confidence:** 3

**Summary:**

Authors propose a novel convolutional architecture for processing volumetric data, based on an orientation invariant convolution operation. The authors give a sound introduction to equivariant/invariant convolutions. The authors propose a method where convolutional kernels are constructed using spherical harmonics, which are subsequently applied to the input signal to obtain orientational feature fields expressed in a basis of circular harmonics. The signal is mapped back to a position-orientation scalar feature map, after which the authors then apply (soft-)max pooling over the orientation axis to map back to the original 3D domain. Authors show results on two datasets of 3D data, very clearly showing a significant reduction in the number of trainable parameters compared to baseline methods.

**Strengths:**

- Good related work section, sketches a clear context for the current work.
- Authors use figures to illustrate their approach.
-The model outperforms baselines significantly with only a fraction of the number of trainable parameters used, indicating the strength of this approach in these experimental settings.

**Weaknesses:**

My main concern is with regards to the clarity of the proposed method.
First, the manuscript seems to lack motivation for your approach. The authors are very thorough in their literature review, and include references and explanation of a lot of relevant equivariant/invariant approaches to convolutions in multiple domains. However, I’m not sure where this method fits in. What are the specific challenges with previous works in invariant/equivariant convolutions applied to 3D data that this method is proposing to alleviate? And how does it compare to previous works using spherical harmonics (e.g. [1, 2, 3]). You mention the need for avoiding a summation over the rotation axis, but I’m not sure where this follows from. Could you expand on this?

From what I gather your work seems like application of convolutions based on spherical harmonics followed by orientational pooling. Neither of these concepts are novel, nor is their combination.  Where is your contribution exactly, and what is its motivation? It seems this approach is severly limiting in the spatial composition of orientational patterns it is able to express, since architectures invariant to rotation are provably less expressive than equivariant architectures (e.g. see fig 1 in [4])

Second, the experiments are somewhat limited. Authors show results only on two datasets, neither of which is very widely carried in the field of learning on 3D data. I would be interested to see performance of your approach on larger-scale data, e.g. non-aligned ShapeNet, ModelNet, QM9.

[1] Thomas, N., Smidt, T., Kearnes, S., Yang, L., Li, L., Kohlhoff, K., & Riley, P. (2018). Tensor field networks: Rotation-and translation-equivariant neural networks for 3d point clouds. arXiv preprint arXiv:1802.08219.
[2] Brandstetter, J., Hesselink, R., van der Pol, E., Bekkers, E. J., & Welling, M. (2021). Geometric and physical quantities improve e (3) equivariant message passing. arXiv preprint arXiv:2110.02905.
[3] Weiler, M., Geiger, M., Welling, M., Boomsma, W., & Cohen, T. S. (2018). 3d steerable cnns: Learning rotationally equivariant features in volumetric data. Advances in Neural Information Processing Systems, 31.
[4] Romero, D., Bekkers, E., Tomczak, J., & Hoogendoorn, M. (2020, November). Attentive group equivariant convolutional networks. In International Conference on Machine Learning (pp. 8188-8199). PMLR.

**Questions:**

What is the motivation for your approach compared to equivariant 3D methods?
What exactly is the contribution of your approach? It seems you are simply applying spherical convolutions followed by orientational (soft-)max pooling? Or am I misunderstanding
Could you relate the steps in your derivation on page 5 to parts of figure 1? I am finding the figure itself somewhat hard to interpret.
Could you add references for the baseline methods you’re comparing against in the experimental section?
Are you planning to provide reference code for your implementation?
I am having a hard time interpreting Figure 2 and the corresponding experiment. What exactly are you trying to illustrate here? How does this toy setting relate to real-world data?
For figure 3, is there any intuition to interpret the visualised filters here? To me, Fig3 and Fig4 show very similar patterns.

---

> ### Author Response · Authors · 2023-11-22
> **Responses to questions**
>
> > **Q1**: What is the motivation for your approach compared to equivariant 3D methods?
>
> Please see the first point of Comments on Weaknesses.
>
> > **Q2**: What exactly is the contribution of your approach? It seems you are simply applying spherical convolutions followed by orientational (soft-)max pooling? Or am I misunderstanding
>
> Yes, you understood it correctly. Please see the detailed explanation above in Comments on Weaknesses. We are convinced that our contribution is that our method meets the assigned requirements that we described in the motivation.
>
>
>
> > **Q3**: Could you relate the steps in your derivation on page 5 to parts of figure 1? I am finding the figure itself somewhat hard to interpret.
>
> We apologize if this seemed unclear.  We have already matched the steps from page 5 with the operations in the figure and numbered them. We will additionally add these operations in the Figure caption for the revision.
> > **Q4**: Could you add references for the baseline methods you’re comparing against in the experimental section?
>
> We thank the referee for the comment. We will add the references to the tables.
>
>
>
> >  **Q5**: Are you planning to provide reference code for your implementation?
>
> We will release the source code in the revision.
>
>
>
> > **Q6**: I am having a hard time interpreting Figure 2 and the corresponding experiment. What exactly are you trying to illustrate here? How does this toy setting relate to real-world data?
>
> We apologize for an unclear explanation. We will make it better in the revision.  Figure 2 demonstrates how the range of sampling maximum values of a function in the *SO(3)* space (with randomly generated coefficients for different orientations) depends  on the sampling size. We aimed to illustrate  that the responsiveness of our convolution output to the changes in the pattern orientation decreases as the size of the (angular) sampling increases. And that even a small sampling size  gives a narrow range of maximum values.
>
>
>
> > **Q7**:  For figure 3, is there any intuition to interpret the visualised filters here? To me, Fig3 and Fig4 show very similar patterns.
>
> In fact, Figure 3 and Figure 4 show different patterns, but they indeed look alike.   The apparent similarity comes from the fact that the angular resolution is quite low since these visualizations correspond to filters that fit into a 3x3x3 cube.  Our main goal was, however, to show a variety of filters and the absence of angular symmetry.

---

> ### Author Response · Authors · 2023-11-22
> **Comments on Weaknesses**
>
> > **W1**: My main concern is with regards to the clarity of the proposed method. First, the manuscript seems to lack motivation for your approach. The authors are very thorough in their literature review, and include references and explanation of a lot of relevant equivariant/invariant approaches to convolutions in multiple domains. However, I’m not sure where this method fits in. What are the specific challenges with previous works in invariant/equivariant convolutions applied to 3D data that this method is proposing to alleviate? And how does it compare to previous works using spherical harmonics (e.g. [1, 2, 3]).
> >  1) Thomas, N., Smidt, T., Kearnes, S., Yang, L., Li, L., Kohlhoff, K., \& Riley, P. (2018). Tensor field networks: Rotation-and translation-equivariant neural networks for 3d point clouds. arXiv preprint arXiv:1802.08219.
> > 2) Brandstetter, J., Hesselink, R., van der Pol, E., Bekkers, E. J., \& Welling, M. (2021). Geometric and physical quantities improve e (3) equivariant message passing. arXiv preprint arXiv:2110.02905.
> > 3) Weiler, M., Geiger, M., Welling, M., Boomsma, W., \& Cohen, T. S. (2018). 3d steerable cnns: Learning rotationally equivariant features in volumetric data. Advances in Neural Information Processing Systems, 31.
>
> Our main motivation was to develop a rotationally-invariant method that
> *  applies to regular/voxelized volumetric data,
> * detects arbitrarily shaped patterns.
>
> Our main contribution is that
> * the method detects arbitrary-shaped filters in contrast to equivariant steerable networks that the reviewer refers to;
> * the method considers continuous space of rotations and avoids summing up in the rotational space, unlike group convolution networks;
> * in contrast to Spherical Convolutions, our method learns joint rotations over concentric shells.
>
>
> >   **W2**:You mention the need for avoiding a summation over the rotation axis, but I’m not sure where this follows from. Could you expand on this?
> > From what I gather your work seems like application of convolutions based on spherical harmonics followed by orientational pooling. Neither of these concepts are novel, nor is their combination. Where is your contribution exactly, and what is its motivation?
>
> We apologize for not being clear in explaining this in the paper. Please see response to Q2 of  the first reviewer to find a detailed explanation.
>
>
>
>
> > **W3**: It seems this approach is severly limiting in the spatial composition of orientational patterns it is able to express, since architectures invariant to rotation are provably less expressive than equivariant architectures (e.g. see fig 1 in [4])
> > 4) Romero, D., Bekkers, E., Tomczak, J., \& Hoogendoorn, M. (2020, November). Attentive group equivariant convolutional networks. In International Conference on Machine Learning (pp. 8188-8199). PMLR.
>
> The referee is absolutely correct, indeed, sometimes taking into account the orientations relation of elements is crucial for spatial data processing. That's why the equivariance with respect to the local patterns orientation is more expressive than invariance. The attention mechanism may provide the equivariance, but it leads to excessive computational overhead when continuous rotations are considered. Another solution might be employing not only scalar but also (hidden) vector  features that we are currently exploring. However, in the ILPO method, we are focused on scalar features and pattern orientation invariance.

---

> ### Author Response · Authors · 2023-11-22
>
> > **W4**:Second, the experiments are somewhat limited. Authors show results only on two datasets, neither of which is very widely carried in the field of learning on 3D data. I would be interested to see performance of your approach on larger-scale data, e.g. non-aligned ShapeNet, ModelNet, QM9.
>
> We were focused on volumetric **regular** data. In fact, in our experiments, only the MedMNIST collection initially contains regular data. We also assessed our method on the CATH dataset to compare our approach with 3D equivariant steerable networks by Weiler et al., 2018.
>  **QM9** contains data on small organic molecules that have a limited number of atoms, up to 9 heavy atoms. It cannot be regarded as regular, and data-specific point cloud-based architectures have already achieved very impressive baselines.
> **ShapeNet** and **ModelNet** can indeed be considered as regular data maps with some resolution.
> However, the original data representation in these cases was in the point-cloud and CAD formats, which are more aptly addressed by the state-of-the-art methods than by a Convolutional Neural Network (CNN). Two main challenges arise when attempting to apply our approach to these datasets. The first challenge is that discretizing the data at a constant resolution results in losing minor details and sharp edges. The second challenge is that representing this data as voxels leads to a sparse configuration. Unfortunately, our method is not suited for handling sparse data mainly due to the polynomial representation of the angular part of the data.

---

### Official Review · Reviewer_KeGD · 2023-11-02

**Soundness:** 3 good
**Presentation:** 3 good
**Contribution:** 3 good
**Rating:** 5
**Confidence:** 3

**Summary:**

This paper proposes a new 3D convolution operation that is invariant to the local orientations of the input features in volumetric data. The method considers rotation invariance over a continuous range of rotations spanning all of SO(3) and demonstrates superior performance on two datasets while being extremely parameter efficient.

**Strengths:**

- The paper is well written, the method is explained comprehensively and is easy to follow.
- The method considers rotation invariance over all of SO(3), and not just a set of discrete rotations as is common in previous work.
- The proposed orientation pooling operation is interesting and leads to expressive filters that not are limited to being radially symmetric.
- The method performs well on the chosen datasets while being extremely parameter efficient.

**Weaknesses:**

- The paper only considers two small datasets with relatively low resolution (50^3 and 28^3) volumetric models. It is not clear how the model performance scales as the resolution of the input increases.
- Some analysis on the computational and memory complexity of the proposed convolution operation, compared to standard convolution, is missing.

**Questions:**

- How does the model perform on datasets with higher resolution data? Considering more diverse geometric datasets like ShapeNet with randomized orientations with higher resolution voxels might help to better demonstrate the generality of the method.
- Does the proposed convolution operation increase the training time compared to standard convolutions? Including some timing or FLOPS comparisons against related work would be helpful.
- Is it possible to achieve rotation sensitivity by making the method equivariant rather than invariant to rotations? Some discussion on this would be useful.
- Typo: Figure 2 x-axis label: sapling -> sampling

---

> ### Author Response · Authors · 2023-11-22
> **Responses to questions**
>
> > **Q1**:  How does the model perform on datasets with higher resolution data? Considering more diverse geometric datasets like ShapeNet with randomized orientations with higher resolution voxels might help to better demonstrate the generality of the method.
>
> In our method, we focused on the invariance to volumetric pattern orientation in regular data. That's why the main part of the experiments was conducted on the MedMNIST collection, where we compared our method with  methods tested on this collection. MedMNIST contains real voxelized regular data, in contrast to ShapeNet and ModelNet, which were designed as point-cloud collections. For these two datasets, the state-of-the-art methods are very data-specific.
> We also aimed to compare our method with invariant methods for regular data (Group convolution and Steerable filters). The former did not show much more efficiency compared to conventional CNNs, unlike the latter, which demonstrated better performance with much fewer parameters.
> Consequently, we were more interested in comparing the ILPO method with 3D steerable networks. Thus, we assessed our method on the CATH dataset on which 3D steerable networks were tested.
>
> > **Q2**: Does the proposed convolution operation increase the training time compared to standard convolutions? Including some timing or FLOPS comparisons against related work would be helpful.
>
> We discussed the complexity in the Appendix. The complexity of the conventional convolution is $O(N^3 L^3 D^2)$, where $N$ is the linear size of the input, $L$ is the linear filter size, and $D$ is the feature space size. For comparison, the complexity of our convolution is $O(N^3 L^6 D^2)$. If we assume that  the ILPO convolution uses $M$ times fewer features than the conventional convolution, and $L^3 << M^2$, then the ILPO convolution has a lower complexity. In our experiments, we satisfied this condition.
>
> > **Q3**: Is it possible to achieve rotation sensitivity by making the method equivariant rather than invariant to rotations? Some discussion on this would be useful.
>
>
> Currently, we are working on an extension of this method that provides equivariance by detecting not only scalar but also multidimensional features. However, we found it impossible to mention this extension in the considered paper because it would require an extensive number of new mathematical notations.
> > **Q4**: Typo: Figure 2 x-axis label: sapling -> sampling
>
> Thank you very much. We will correct it.

---

> > ### Comment · Reviewer_KeGD · 2023-11-23
> >
> > Thanks for the responses, my questions have been largely clarified.

---

> ### Author Response · Authors · 2023-11-22
> **Comments on Weaknesses**
>
> > **W1**: The paper only considers two small datasets with relatively low resolution (50^3 and 28^3) volumetric models. It is not clear how the model performance scales as the resolution of the input increases.
>
> We believed that tests on the datasets used would be sufficient to prove our concept. Moreover, these datasets contained  various volumetric patterns.
>
> > **W2**: Some analysis on the computational and memory complexity of the proposed convolution operation, compared to standard convolution, is missing.
>
> Thank you for your comment. We analyzed computational complexity in the Apppendix. We will add an analysis of memory complexity for the revision.

---

### Official Review · Reviewer_PtnA · 2023-11-03

**Soundness:** 2 fair
**Presentation:** 2 fair
**Contribution:** 2 fair
**Rating:** 5
**Confidence:** 2

**Summary:**

This paper designs a rotation-invariant 3D convolutional neural network architecture. Unlike prior works that rely on data augmentation to learn rotation-invariant representation, the proposed method incorporates such invariance into the model, which leads to a more lightweight and efficient network architecture. Experiments on two volumetric datasets show the effectiveness of the proposed method over several baselines that rely on data augmentation.

**Strengths:**

1. The proposed method is technically sound to make the 3D ConvNet rotation-invariant by design.
2. This paper is overall easy to follow.
3. Experiments on two datasets show the proposed method achieves similar or better performance with much less computational cost, in comparison with baselines that rely on data augmentation.

**Weaknesses:**

My main concern with this work is the lack of comparisons with existing methods. This paper mainly makes comparisons with ResNet variants that learn rotation-invariant representations via data augmentation. How does it compare to prior works that incorporate the rotation-invariance in the network design such as CubeNet? What is the difference between this work and prior works (such as https://arxiv.org/pdf/2003.08890.pdf)?

**Questions:**

See the weakness section.

---

> ### Author Response · Authors · 2023-11-22
> **Comments on Weaknesses**
>
> We thank the reviewer for the valuable comment and apologize for not being clear enough in the article. We  mentioned all these methods in the Related Work section and explained  the differences and advantages of our technique. We will make it more apparent in the revision. The difference between ILPO and the group convolution methods is that we consider continuous space of rotations. The last method is indeed similar to our approach, but the authors restricted the shape of the applied filters (they considered steerable filters that are not arbitrary). Moreover, the method was tested on datasets where patterns have symmetries (crosses in the first dataset and spherical lung nodules in the second one). Additionally, we optimized the workflow that removes sampling in the rotation space from the complexity of the convolution operation.

---

### Official Review · Reviewer_sNRn · 2023-11-05

**Soundness:** 2 fair
**Presentation:** 2 fair
**Contribution:** 3 good
**Rating:** 5
**Confidence:** 4

**Summary:**

In this paper the authors propose a convolutional neural network equivariant to local 3D rotational deformations. To do this the authors represent filters in their spherical harmonic decomposition. The authors show the proposed approach achieves better performance than baseline models, even with a significant reduction in the number of learnable parameters.

**Strengths:**

*Originality:* The proposed approach appears novel.

*Quality:* In the experiments, proposed method performs on-par with or better than most baselines.

*Clarity:* (see weakness)

*Significance:* Designing locally equivariant neural network models is a challenging and important problem in the field.

**Weaknesses:**

*Quality/Clarity:* The presentation of the related work, and the critiques therein are a bit confusing given the proposed approach.
* The related work is divided into (1) “those based on equivariant operations in the SO(3) space” and (2) “those with learnable filters that are orthogonal to the SO(3).” If I understand correctly, it seems the categories would be better named (1) finite group convolution, and (2) continuous group convolution methods.
* The authors describe the methods in (1) as limited in that they require averaging over the rotations, however, this is a consequence of the definition of convolution. Perhaps this was supposed to be a comment on the pooling layers?
* The authors describe the methods in (2)  as limited in that they require inputs to be centered. My understanding is that inputs to these classes of CNNs are centered by mean subtraction, and that translation offsets do not limit their applicability.
* The authors critique methods in the second class saying “[these methods use] equivariant quantities by design, which prevents them from capturing complex patterns involving multiple point” which is strange since the proposed method requires the filters to be learned as linear combinations of Wigner-D matrices, making them equivariant by design…
* As I understad, one of the claims of the paper is that it can be applied to SE(3) data, but it doesn’t seem like that was tested in the experiments
* It is difficult to evaluate the quality of the model since the experiments are quite sparse. Only one of the related works are compared against and only in a single setting.
* Some notation is introduced without explanation (see questions)

**Questions:**

*Questions:*
- What is $\Omega_r$ in eqn (13 & 15)
- It isn’t obvious to me why (16) would be more expressive than averaging especially since it seems like it is the same for all R, can the authors give some intuition here

*Possible typos:*
- “architecture” level → ``architecture” level
- In Equation (1 & 2) f(r + \Delta) should be f(\Delta - r)

---

> ### Author Response · Authors · 2023-11-22
> **Responses to questions**
>
> > **Q1**:  What is $\Omega_r$ in eqn (13 & 15)
>
>
> We apologize for not being clear, $\Omega_r$ is the angular part of the vector $\vec r$ in $3D$.
>
> > **Q2**: It isn’t obvious to me why (16) would be more expressive than averaging especially since it seems like it is the same for all R, can the authors give some intuition here
>
> Averaging (or summing) of a function in $3D$ annihilates angular dependencies of a filter. Let's consider a filter defined by a function $g(\vec{r})$:
>     \begin{equation}
>         g(\vec{r}) = \sum_{l = 0}^{L-1} g^k_l(r) Y^k_l(\Omega_r),
>     \end{equation}
>     where $(r, \Omega_r)$ are the radial and the angular components of the vector $\vec{r}$, and $g^k_l$  are the spherical harmonic expansion coefficients of a function $g(\vec{r})$.
>     Then we rotate this function by
>     $\mathcal{R} \in \text{SO(3)} $
>     and convolve with $f(\vec{r})$:
>     \begin{equation}
>         h(\vec{\Delta}, \mathcal{R}) = \int_{\mathbb{R}^3}  f(\vec{\Delta} - \vec{r}) g(\vec{r}) d \vec{r}
>     \end{equation}
>     If we integrate this result over all rotations in SO(3),
>     which is approximately equal to summing the function over a finite set of (equally distributed) rotations,
>     %that is analogical to summation:
>     we obtain
>     \begin{equation}
>         h(\vec{\Delta}) = \int_{\text{SO(3)}}  h(\vec{\Delta}, \mathcal{R}) d \mathcal{R} = 8 \pi^2 \int_{\mathbb{R}^3}  f(\vec{\Delta} - \vec{r}) g^0_0(r) d \vec{r},
>     \end{equation}
>     where $ g^0_0(r)$ are the zero-order expansion coefficients that equal to the mean value of the integrated function over the domain.
>     Thus, we can conclude that summing over all rotations in SO(3) of the result of the convolution with an arbitrary filter is equivalent to a convolution with a radially-symmetric filter.
> On the contrary, equation (16) allows us to keep the dependency of the convolution result on the filter's orientation.

---

> ### Author Response · Authors · 2023-11-22
> **Comments on Weaknesses**
>
> >  **W1**:  The related work is divided into (1) “those based on equivariant operations in the SO(3) space” and (2) “those with learnable filters that are orthogonal to the SO(3).” If I understand correctly, it seems the categories would be better named (1) finite group convolution, and (2) continuous group convolution methods.
>
> In principle, we can also group methods into finite and continuous group convolutions, but in the present classification, the first class can already contain both. The second one considers *continuous equivariant filters* by design.
>
> > **W2**: The authors describe the methods in (1) as limited in that they require averaging over the rotations, however, this is a consequence of the definition of convolution. Perhaps this was supposed to be a comment on the pooling layers?
>
> We agree that "averaging" over rotations is a consequence of the definition of the convolution. We just wanted to point out that what may be an advantage in recognizing patterns on a sphere turns out to be a disadvantage in detecting volumetric patterns. Averaging over the rotations of the 3D convolution is identical to averaging the filter over all rotations before the convolution.
>
>
>
> > **W3**:  The authors describe the methods in (2) as limited in that they require inputs to be centered. My understanding is that inputs to these classes of CNNs are centered by mean subtraction, and that translation offsets do not limit their applicability.
>
> We thank the reviewer, it was our fault not describing it clearly, and we will make corrections in the revision. In the text, we wanted to say  that the convolution on $S^2$ and SO(3) don't allow to detect patterns in $R^3$ without further extensions of the methods. Indeed, if the volumetric data is represented as a set of concentric spherical shells, the methods focused on data in  $S^2$ and SO(3) don't provide the opportunity to detect patterns lying between the spheres. Or, in other words, data and patterns on different concentric shells are learned *independently*, which reduces the expressivity of the methods.
>
>
>
>
>
>
>
>
> > **W4**:  The authors critique methods in the second class saying “[these methods use] equivariant quantities by design, which prevents them from capturing complex patterns involving multiple point” which is strange since the proposed method requires the filters to be learned as linear combinations of Wigner-D matrices, making them equivariant by design…
>
> Again, we were not very clear. In this sentence, we wrote about EGNN that uses pairwise distances and internode vectors as equivariant features. We will correct it in the revision.
>
>
>
>
>
> > **W5**:  As I understad, one of the claims of the paper is that it can be applied to SE(3) data, but it doesn’t seem like that was tested in the experiments
>
>   We apologize that we didn't specify it explicitly in the text. The fact is that we assessed our model on 3D data in which volumetric patterns occur repeatedly, oriented arbitrarily and located in different spots of the input images.
>
>
>
> > **W6**:  It is difficult to evaluate the quality of the model since the experiments are quite sparse. Only one of the related works are compared against and only in a single setting.
>
> We conducted experiments for two different datasets (the second dataset is factually a collection of datasets) where pattern recognition is quite challenging due to arbitrary rotation and translation of the volumetric images.
>         The first dataset is built around the CATH classification problem, where we compared our method with the most popular equivariant method for regular volumetric data.
>         The second part of the experiments was conducted with the MedMNIST collection, where we considered all published methods that were assessed on this collection.
>         For both parts of the experiments, we used equivariant/invariant methods to compare. In the CATH experiments, such a method was the **3D steerable networks**. For the second part, we compared our method with **SE3MovFrNet**, which is invariant and was mentioned in the section "Related Work".  We compare our method with the settings of other methods presented by the methods' authors.
>
>
>
>
>
> > **W7**:  Some notation is introduced without explanation (see questions)
>
>   We thank the reviewer for pointing this out, we have corrected all the notations for the revision.

---

### Meta-Review · Area_Chair_AYrk · 2023-12-15

**Metareview:**

The paper achieves rotational invariance for 3D volumetric data with a combination of spherical convolutions and orientational max pooling and shows superiority of soft max pooling. It achieves improved results on several MedMNIST 3D data benchmarks compared to augmentation strategies as well as some baseline methods such as steerable ResNet.


Reviewers have presented thorough reviews which include several concerns, to which the authors have provided rebuttals. While the AC agrees with the importance of the problem, and the relevance of the proposed method, the paper requires a major revision to increase its presentation and relevance. The AC, thus, recommends rejection.


Among the many points raised by the reviewers that should be addressed for a next version, the AC believes the contributions compared to prior works should be clearly specified and the results thoroughly compared with the many existing prior works.

**Justification For Why Not Higher Score:**

The paper seems to be in early-mid stage of progress as well as writing.  It requires to better flesh out its contributions, solidify the evidence for its efficacy, and present them in a convincing manner.

**Justification For Why Not Lower Score:**

N/A

---

### Decision · Program_Chairs · 2024-01-16

Reject